# Research on Consumer Perception Regarding Wine Products and Wine Tourism in the Republic of Moldova

Viorica Guțan [1,2], Larisa Șavga [1,2], Constanta Laura Zugravu [1,*], Diana Bucur [2] and Gheorghe Adrian Zugravu [1]

[1] Faculty of Engineering and Agronomy in Braila, "Dunarea de Jos" University of Galati, 800008 Galaţi, Romania
[2] Faculty of Economic and Legal Sciences, Trade Co-Operative University of Moldova, Gagarin, 8, MD 2001 Chisinau, Moldova
* Correspondence: laura.zugravu@ugal.ro

**Abstract:** Traditions of vine cultivation and wine production have been formed in Moldova over centuries. According to folk traditions, wine is an integral part of any event organized by the locals. Wine tourism is a particularly significant sector for the country's economy. Although it only emerged at the end of the 20th century, it is growing in importance every year. To contribute to this area of scientific discourse, a study on consumers' perception of wine products and wine tourism in the Republic of Moldova was conducted. The main respondents who participated in the survey were citizens of the country, but respondents from Romania and Ukraine also participated. The results obtained indicate that wine products are consumed by the majority of the respondents participating in the survey and that wine tourism has continuity in its development and is of clear interest among citizens. The diversity of the tourist offers of the wineries and the recreational areas in which they are located are of course of particular importance when selecting a wine tour.

**Keywords:** respondents; questionnaire; consumers; visitors; wine product; wine tourism





## 1. Introduction

The world of wine is divided into two large world areas: the "Old World" with France, Italy, Spain, Germany, followed by Austria, Hungary, Slovenia, Croatia, Greece and the "New World" with Canada, the United States, Chile, Argentina, South Africa, Australia and New Zealand [1–3]. The successful model and trend of food and wine tourism adopted above all from the United States is nothing more than the result of inspiration from the European heritage with the attempt to reproduce the same atmosphere as in Les Routes des Vins of Bordeaux, the domains of Burgundy, and the wine and flavor routes of Italy [4,5]. Rural areas cover most of Europe's surface. The urban-rural relationship has always been a characteristic feature of the European area leading to a permanent profitable exchange of goods, people, resources, knowledge and lifestyles. Although in the second half of the 20th century this relationship temporarily entered a crisis, there is no doubt that the rural natural—human landscape represents a factor of strong territorial identity and a resource of primary importance, both for local communities and for the community. Moreover, wine tourism represents one of the levers to tie the threads of the dialogue between the city and the countryside. The wine from the territories where it is produced is the pivot around which revolve a multiplicity of consumption activities. These generate income, jobs, economic and, consequently, social value. It should be noted in this respect that 50% of Italian wine production still belongs to a specific place. In recent decades, however, micro-enterprises have decreased. In 2015, there were 34,166 companies that produced less than 100 hL, representing 74% of the total [6–8].

One of the main interests of the citizens of modern society is the organization of their free time. In order to cover this demand with an incredible speed and a remarkable continuity of growth, the tourism industry has developed in the last century, both worldwide

and in the Republic of Moldova [9–11]. Tourism has proven to be a very important activity for the economy of our country. Tourism in the Republic of Moldova has a rather large potential for development, presenting a complex combination of natural environments (natural areas, massive forests in the center of the country, rocks, valleys, meadows) and man-made ones (medieval settlements, fortresses, cities with various architectural styles, spas, festivals, wineries, etc.) [12–15].

Our research into consumer perception regarding wine products and wine tourism is concentrated on representative wineries from the Republic of Moldova (Figure 1).

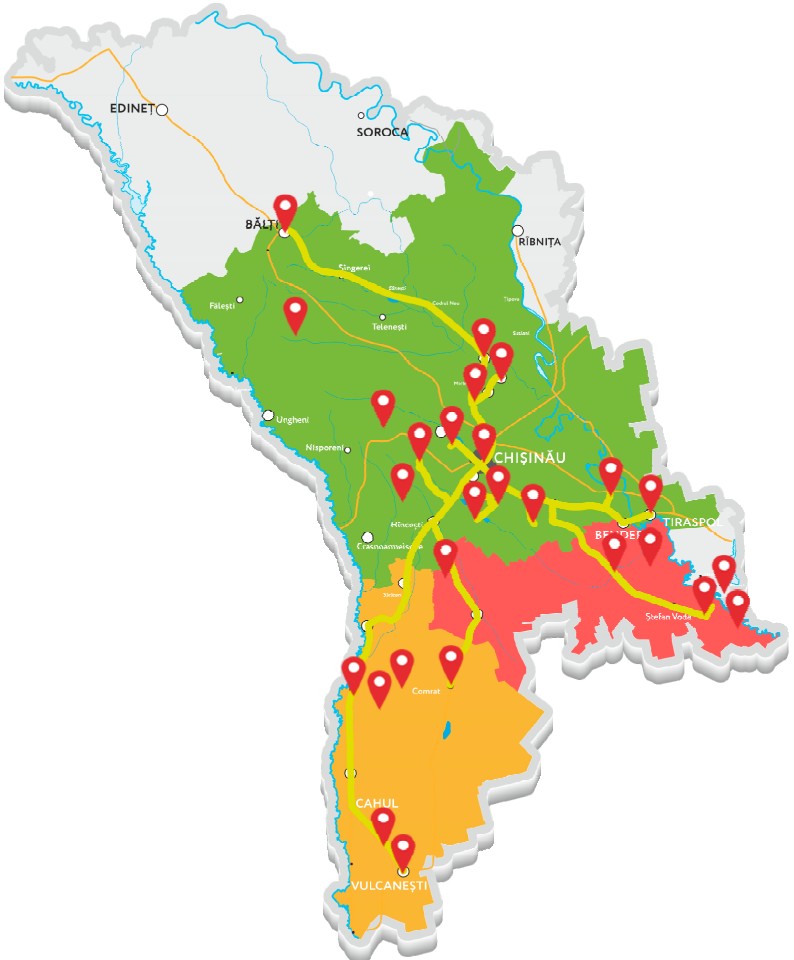

**Figure 1.** Map of wineries in Republic of Moldova. Sources: https://wineofmoldova.com/en/ (accessed on 16 March 2023).

Among citizens there is a growing demand for short-term vacations, weekend vacations, and individualized tourism. Today's tourists show a growing interest in tranquility, direct contact with nature, culture and avoiding crowds [16–18]. At the same time, tourists advocate for relaxing tourist visits that stimulate them mentally and physically. In our opinion, wine tourism could satisfy the stated requirements. It is considered to be one of the forms of tourism of the Republic of Moldova which, through its potential, could easily compete on the international market [19–21].

The purpose of this research is to analyze consumers' perception of wine products and wine tourism, which can thus support the foundation of a wine tourism development strategy in the Republic of Moldova, starting from the identification and validation of the factors, expressed in hypotheses, which can stimulate the consumption of wine products.

Starting from the concepts found in the literature with the help of the bibliometric study carried out using VOSviewer software, we created a questionnaire to investigate the perception of consumers of wine products and wine tourism in the Republic of Moldova.

At the same time, with the help of the questionnaire, we also analyzed the importance of the main factors that can contribute to the development of the wine sector in the Republic of Moldova. In this sense, with the help of a fuzzy application developed in Matlab, we have developed a qualitative fuzzy analysis methodology. We thus managed to identify the importance of the factors that can stimulate the development of this tourism sector in the Republic of Moldova.

## 2. Literature Review

The art of winemaking has been valued by the population of the country since ancient times. In the country's aquatic spa treatment centers, grapes and grape juice are widely consumed [22,23]. Wine-related tourism offers a variety of guided tours in wine cellars, in underground cities with streets named after grape varieties, in underground galleries, in primary wine processing enterprises, in the production of sparkling wine, divine balsams, in wine cellars, in bottling rooms, rooms with a variety of themes, wine tasting rooms, banquet rooms, conference rooms, in wine shops, etc. [15,24,25]. Winery visitors, in addition to the basic program of guided tours and tastings, could also benefit from other services. These include: custom wine orders, tours of vineyards and other nearby attractions, bike tours, picnic areas, hotel services, terraces, children's playgrounds, museums, art galleries, fishing, hunting, swimming pools, saunas, culinary master-classes, etc. [26–30]. During the season, tourists can participate in the wine production process. Most wine companies offer the possibility of organized tours for visitors [31–36]. Here tourists can gain experience and learn about the complex production process, be present at the bottling process and, of course, taste the final product [37–40].

At a higher level are the generic wines with the possibility of indicating vintage and/or variety but lacking indications of origin; these can be produced with grapes from various areas and/or from various EU Member States. In contrast, the Protected Geographical Indication based on the name of a region or a specific place serves to designate a wine—originating in that region or place of a certain quality, reputation or other characteristic which may be attributed to the geographical origin. At least 85% of the grapes from which a PGI wine is obtained must come from that grape's geographical area [41–43].

In Moldova there are tourist routes "Wine Road in the Republic of Moldova" and "Wine Road of Moldova", which represent a substantial reason to visit the Republic of Moldova [44–48].

In recent years, thanks to the investments of the winery owners, and the support through grants and assistance of the Moldova Competitiveness Project, financed by USAID and the Government of Sweden, the number of wineries open to tourists has doubled [49].

## 3. Methodology

In order to investigate the stated theme, the authors used a series of research methods: exposition, systematization, questionnaire investigation, induction and deduction, comparison, quantitative and qualitative analysis, synthesis. The questionnaire survey is a method of questioning social facts (opinions, attitudes, motivations, etc.) at the level of human groups, smaller or larger, and of analyzing quantifiable data in order to describe and explain them [50–52]. The survey is a fast and efficient method of describing opinions. On the other hand, esearch studies also make use of complementary methods (observation, analysis, documentation) [53–57]. The survey on consumer perception of wine products and wine tourism in the Republic of Moldova was conducted online on Google forms, mainly among the citizens of the Republic of Moldova, using a sample of approximately 110 people. A total of 200 questionnaires were distributed in 32 counties and 134 were returned. After checking the validity of the returned questionnaires, 24 questionnaires that were incomplete or had logical mistakes were deleted, meaning that 110 valid questionnaires were obtained; the effective response rate was 55%.



The sample for this population was drawn randomly by the RDD method (random digit dialing) and was achieved by calculating the weights depending on the region and the environment of residence and gender.

Given that representativeness depends on the sample generated by means of statistical tools, eventually, there may be no exact relationship between the concepts of representativeness and coverage, since the latter is the proportion of elements selected in the sample with respect to the population.

The research results allowed making some remarks and drawing some conclusions regarding consumer perception of wine products and the development of wine tourism.

The research results presented here were derived using the graphic and numerical methods. The graphical method is used to visually identify the trends in the data. The following types of graphs were used in the present work: bar graph, line graph, pie graph and diagram. The numerical data are more objective and accurate. Since they complement each other, it is good to use these methods in combination.

The synthesis and deduction methods were used to formulate conclusions and argue for some recommendations.

Furthermore, through inductive reasoning, data collected during semi-structured interviews are linked to relevant theories gathered during the systematic literature review using VOSViewer 1.6.19 software.

The first objective of this study was to carry out a bibliometric study of wine tourism products. The research of the database of ISI indexed articles, related to the traditional products, was done for ten years (2012–2022) on the trends regarding the traditional product concept. The literature was extracted and analyzed using the Web of Science database. VOSViewer software was used to identify and visualize key trends, influential authors and journals.

Furthermore, 340 WOS articles from 2019 to 2023 were selected for detailed study based on three main criteria:

- Topics regarding the wine products and wine tourism,
- Document type "article"
- Year of publication in the period 2019–2023.

We performed several types of analyses regarding wine and wine tourism products; we analyzed the period 2019–2023 in its entirety, after which we analyzed the most cited articles and the most recently accessed articles. Extracting information from the Web of Science Database and VOSViewer software were the main techniques for analysis and reporting. Research themes were extracted and analyzed to identify and visualize main trends, (influential) authors and related journals.

VOSviewer provides visualizations of bibliometric networks. VOSviewer therefore displays only the nodes in a bibliometric network and does not display edges between nodes. In the visualizations provided by VOSviewer, the distance between two nodes roughly indicates the relationship between the nodes. By providing distance-based rather than graph-based visualizations, VOSviewer is particularly suitable for viewing larger networks. Because of its strong focus on visualization, VOSviewer offers less functionality for bibliometric network analysis than other tools. However, VOSviewer has some special text extraction features.

We applied VOSviewer to data obtained from the Web of Science database as early as 2019 to ensure that only hot topics are included to detect trends in wine tourism. We did not focus only on the top journals in this field, but used a keyword approach applied regardless of the source of the paper. Thus, brand management has been investigated using inclusion criteria based on thematic keywords found in more than twelve sources. Based on this, grouping could be developed. There are six major keyword clusters related to wine tourism in the most cited WOS articles from 2019–2023 (Figure 2), which we determined based on thematic clusters as:

- "wine tourism";
- "strategy";

- "rural tourism";
- "regions";
- "dimensions";
- "wine routes".

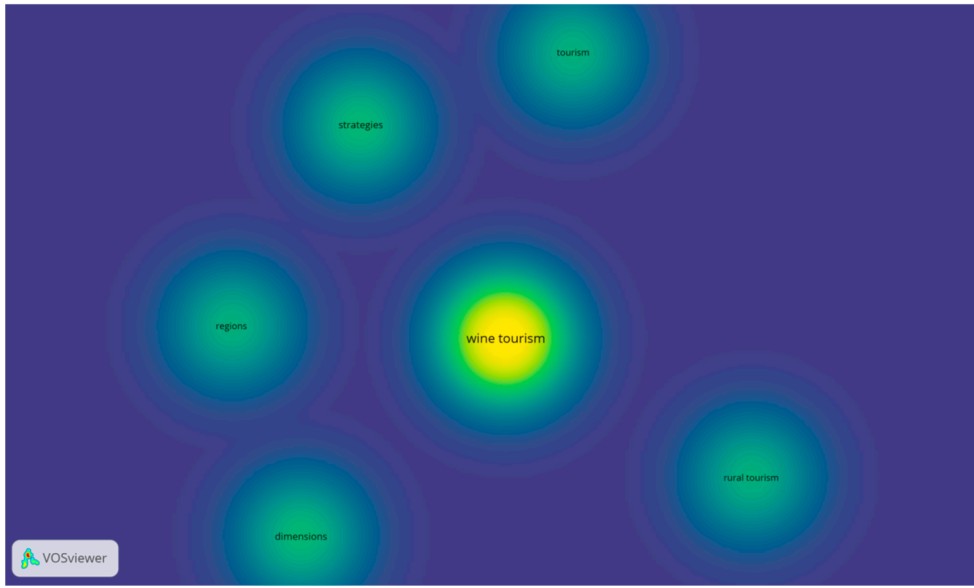

**Figure 2.** Major keyword clusters related to wine tourism in the most cited WOS articles from 2019–2023.

In this scientific investigation the authors present the results of a social survey. It was carried out in order to discover the preferences of different wine consumers and elucidate the interest of the population that participated in the survey regarding the visit to wineries in the Republic of Moldova and the degree of satisfaction received as a result of these visits [50,58–60]. The exposed results can be seen in Figure 3.

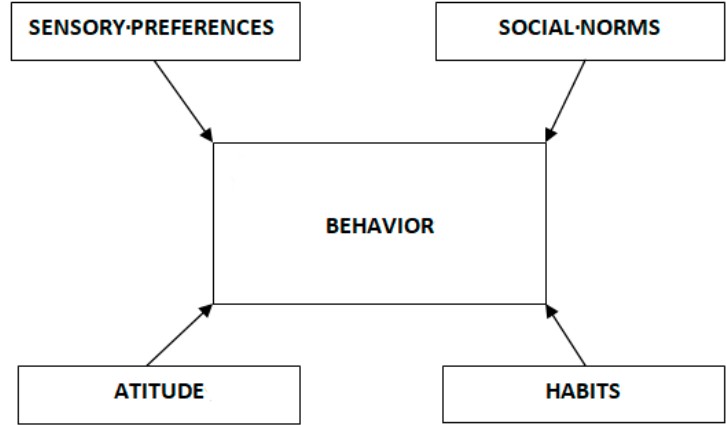

**Figure 3.** The design of research hypotheses for a social survey of wine tourism consumers.

Based on the conceptual model, the following research hypotheses were established:

**Hypothesis 1.** *There is a significant relationship between consumer attitude and intention to purchase wine tourism products.*

**Hypothesis 2.** *There is a significant relationship between social norms and the consumer's intention to purchase wine tourism products.*

**Hypothesis 3.** *There is a significant relationship between consumer habits and consumer intention to purchase wine tourism products.*

**Hypothesis 4.** *There is a significant relationship between sensory preferences and consumer intention to purchase wine tourism products.*

These research hypotheses on the factors that influence the behavior of consumers of wine products were verified with the help of the questionnaire, as described below.

## 4. Results

The survey on consumer perception of wine products and wine tourism in the Republic of Moldova included 20 questions. The first 5 questions of the survey included general data on the participants. These included: citizenship, place of birth, gender, age, and highest level of education of survey respondents. Questions 6–13 of the survey include data on consumption of wine products, their characteristics, disclosure of product information, reason for consumption, frequency of consumption, etc. Survey topics 14–20 include questions regarding visits to wineries and respondents' impressions of the results of these trips.

According to the data collected from the survey, citizens of the Republic of Moldova participated in the proportion of 96.3%, while Romanian citizens and citizens of other states have an equal share in the proportion of 1.8%. They are mainly from the academic community, but also from different branches of the national economy. Of the total number of respondents, 50.9% were born in rural areas and respectively 49.1% were born in urban areas. Respondents were 85.3% women and 14.7% men.

The age of the respondents who took part in the survey varied widely and can be seen in Table 1. The analysis of consumer perception used as segmentation criteria: age, gender and education according to Table 1.

**Table 1.** Distribution of respondents by age.

| Demographic Variables | Categories | Subjects No. | Percent % |
|---|---|---|---|
| Gender | Male | 16 | 14.70 |
| | Female | 94 | 85.30 |
| Age | 18–20 | 11 | 10.1 |
| | 21–30 | 22 | 20.2 |
| | 31–40 | 16 | 14.6 |
| | 41–50 | 33 | 30.3 |
| | 51–60 | 17 | 15.6 |
| | >60 | 10 | 9.2 |
| Place of birth | Rural | 56 | 50.9 |
| | Urban | 54 | 49.1 |

Table 1 shows the proportion of respondents who participated in the survey according to their age. From the graph data, we notice that respondents of different ages took part in the survey: young, middle-aged and elderly people. Out of 110 responses regarding age, the majority of respondents, 33 people or 30.3%, are aged 41–50, followed by respondents aged 21–30 in second place (22 people or 20.2%); in 3rd place are the respondents aged 51–60 with a share of 15.6% (17 people), closely followed by those aged 31–40 years with a share of 14.6% (16 people). Finally, in the last place, we find the respondents who have reached the age of 18–20 years with a share of 10.1% (11 persons) and those aged over 60 years with a share of 9.2% or 10 persons.

We conclude that the participation of different age groups in the survey will allow expressing the most truthful answers to the questions in the questionnaire.

When asked about the last level of education, we establish that respondents with university studies reach a share of 45.9%, those with postgraduate studies (master's, doctorate, etc.) reach a share of 36.7%, those with high school and post-secondary studies

reach a share of 7.3% each, and respondents without high school studies reach a share of 2.8%. Thus, the majority of respondents in the proportion of 82.6% are respondents with university and post-university education.

The investigation of the number of people who consume wine products showed that 60.9% of the respondents are regular consumers of these products, 30.9% consume wine products sometimes or rarely and 8.2% do not consume or are not interested in consuming these products. Therefore, out of the total number of respondents surveyed, 91.8% are consumers of wine products.

The next research question generated multiple responses regarding the sources/channels of information about wine products that are most accessible to the respondents. A total of 112 responses were obtained with the following results: media (television, press, etc.) 30.4% (34 people); internet 32.1% (36 people); store shelves 38.4% (43 people); specialized stores 36.6% (41 people); national days, festivals, exhibitions 40.2% (45 people); friends/colleagues 36.6% (41 people); other sources 5.4% (6 people). Thus, we observe that all usual sources of information more or less promote the consumption of wine products.

Multiple responses were also found to the question concerning the place where wine products are more often available to consumers for purchase. Consequently, the following results were obtained concerning the source of wine products: from own household 29.9% (23 persons); from specialized shops 50.9% (56 persons); directly from the factory 9.1% (10 persons); from supermarkets, markets, small shops 51.8% (57 persons); from fairs organized for special occasions 27.3% (30 persons); from festivals 16.4% (18 persons); from exhibitions 25.5% (28 persons); from tourist reception structures with public catering functions 2.7% (3 persons); from other places 3.6% (4 persons). Thus, wine consumers most often purchase wine products from specialized shops and markets.

Another research question concerned the reason for the choice of a wine product made by the winemakers of the Republic of Moldova. The ratings of the motives were: very important, important, indifferent, less important, slightly important. Thus, the motive of patriotism was rated as very important by 22%, important by 42%, indifferent by 8%, less important by 6% and slightly important by 8%. The motive of habit passed on by parents and grandparents was rated as very important by 6%, important by 31%, indifferent by 21%, less important by 7%, and slightly important by 9%. The motive of recently acquired habit was rated as very important by 5%, important by 43%, indifferent by 15%, less important by 11%, and slightly important by 2%. The motive of another person's suggestion was rated as very important by 10%, important by 47%, indifferent by 12%, less important by 6%, and slightly important by 6%. The motive of mere curiosity was rated as very important by 9%, important by 23%, indifferent by 28%, less important by 4%, and slightly important by 7%. The motive of promotion via different channels was rated as very important by 47%, important by 26%, indifferent by 6%, less important by 1%, and slightly important by 1%. The motive of food safety was rated as very important by 61%, important by 18%, indifferent by 4%, and slightly important by 2%. The motive of product characteristics (appearance, exterior, color, smell, taste, etc.) was rated as very important by 52%, important 27%, indifferent 7%, less important 1% and slightly important 3%. The curative motive (beneficial to health in small amounts) was rated as very important by 36%, important by 42%, indifferent by 9%, less important by 1%, and slightly important by 2%.

Thus, most of the motives presented in the questionnaire were rated as very important and important.

To analyze the working hypotheses, we processed the answers obtained with the help of the online questionnaire by means of a fuzzy analysis model of the hypotheses that influence the behavior of consumers of wine tourism products. For this purpose, we used the set of tools offered by Matlab R2020b.

Thus, for the analysis of the 4 hypotheses (sensory preference, social norms, attitudes, habits) we calculated an index regarding the importance of each of the working hypotheses in the manifestation of consumption behavior. Based on the data provided by the respondents, we converted the linguistic qualitative assessments (unsatisfactory, satisfactory,

average, good, excellent) into a vector of 3 numerical values (fuzzy triplet), because we used a triangular membership function.

This set of three numerical values (fuzzy triplet) is used to reflect the relative importance of the working assumptions.

$$IH1 = CA/Qt \tag{1}$$

$$IH2 = CSN/Qt \tag{2}$$

$$IH3 = CH/Qt \tag{3}$$

$$IH4 = CSP/Qt \tag{4}$$

where:

CA—the set of three numerical values of the attitude belonging to three of the 5 linguistic terms of the quality assessment scale,

CSN—the set of three numerical values of social norms belonging to three of the 5 linguistic terms of the quality assessment scale,

CH—the set of three numerical values of habits belonging to three of the 5 linguistic terms of the quality assessment scale,

CSP—the set of three numerical values of the sensory preferences aspect belonging to three of the 5 linguistic terms of the quality assessment scale.

For the defuzzification of global indices of sensory quality, we used a function created in Matlab called df.m

%defuzzificare

$$\text{function } Y = df(A)$$

$$Y = (3 \times A(1) - A(2) + A(3))/3$$

To order the importance indices of each hypothesis (obtained after defuzzification) and to determine the position within this comparative analysis, the following relationships are used in Matlab (according to expressions (1)–(4), and Tables 2–5):

$$CS = [df(IH1)\ df(IH2)\ df(IH3)\ df(IH4)];$$

$$[CSd, Ld] = sortrows(CS', -1);$$

$$[La, L] = sortrows(Ld);$$

**Table 2.** The importance weight ofe each hypothesis.

| Hypotheses | Unimportant | Little Importance | Important | Very Important | Extremely Important |
|:---:|:---:|:---:|:---:|:---:|:---:|
| H1 | 0 | 0 | 33 | 44 | 33 |
| H2 | 0 | 0 | 18 | 59 | 33 |
| H3 | 0 | 0 | 22 | 55 | 33 |
| H4 | 0 | 0 | 33 | 44 | 33 |

The 10th research question determined what kind of wine products are consumed by the interviewed respondents. Thus, 86% of the respondents prefer to consume moderately alcoholic beverages (wines, sparkling wines), and 14% of the respondents prefer to consume strong spirits (vodka, brandy, cognac, whiskey, liqueur, etc.). Thus, the majority of respondents prefer moderately alcoholic drinks.

On the basis of the 11th research question, the type of wine product consumed predominantly by the surveyed consumers was determined. Thus, 50% of the respondents consume wine, 30.9% consume sparkling wine, 12.7% consume vodka, brandy, cognac, whiskey, 2.5% consume liquor and 4% other beverages.

**Table 3.** Calculation of the triplets associated with the weight of the importance of each hypothesis.

| Hypotheses | Calculation of the Set of Numerical Values Associated with the Quality Attributes, in Matlab Associated with the Importance of Each Hypothesis | Triplets Associated with the Importance of Each Hypothesis | | |
|---|---|---|---|---|
| H1 | CA = (0 × [0 0 25] + 0 × [25 25 25] + 6 × [50 25 25] + 8 × [75 25 25] + 6 × [100 25 0])/110 | 75 | 25 | 17.5 |
| H2 | CSN = (0 × [0 0 25] + 0 × [25 25 25] + 3 × [50 25 25] + 11 × [75 25 25] + 6 × [100 25 0])/110 | 78.75 | 25 | 17.5 |
| H3 | CH = (0 × [0 0 25] + 0 × [25 25 25] + 4 × [50 25 25] + 10 × [75 25 25] + 6 × [100 25 0])/110 | 77.5 | 25 | 17.5 |
| H4 | CSP = (0 × [0 0 25] + 0 × [25 25 25] + 6 × [50 25 25] + 8 × [75 25 25] + 6 × [100 25 0])/110 | 75 | 25 | 17.5 |
| | Qt = CA(1) + CSN(1) + CH(1) + QT(1) + CSP(1); | | 382.5 | |

**Table 4.** Calculation of the triplets associated with the weight of the importance index of each hypothesis.

| Hypotheses | Calculation of the Set of Numerical Values Associated with the Quality Attributes, in Matlab, Associated with the Importance Index of Each Hypothesis | Triplets Associated with the Importance Index of Each Hypothesis | | |
|---|---|---|---|---|
| H1 | IH1 = QC/Qt | 0.1961 | 0.0654 | 0.0458 |
| H2 | IH2 = QA/Qt | 0.2059 | 0.0654 | 0.0458 |
| H3 | IH3 = QG/Qt | 0.2026 | 0.0654 | 0.0458 |
| H4 | IH4 = QT/Qt | 0.1961 | 0.0654 | 0.0458 |

**Table 5.** The place obtained by the index of the importance of each hypothesis in the comparative analysis.

| The Value Associated with the Importance of Each Hypothesis Index, CS | The Value Associated with the Importance of Each Hypothesis Index, in Descending Order, CSd | The Initial Position in the Comparative Analysis |
|---|---|---|
| CS1 = 70.8170 | CSd1 = CS4 = 71.3521 | 4 |
| CS3 = 68.7418 | CSd2 = CS1 = 70.8170 | 1 |
| CS3 = 68.7418 | CSd3 = CS3 = 68.7418 | 3 |
| CS4 = 71.3521 | CSd4 = CS3 = 68.7418 | 2 |

Based on the 12th question in the questionnaire, the criteria for choosing a wine product for consumption were researched. These covered: region/area of origin, manufacturer's logo, Protected Geographical Indication inscription, inscription on the product, familiarity with the product, type of dishes with which it is consumed, quality of the product (color, taste, smell, internal and/or external appearance), the attractiveness of the packaging, price of the product. There were 5 ratings used: very important, important, indifferent, less important, and slightly important. The maximum scores for the criteria presented were obtained by the very important and important qualifiers. According to the responses, the criterion of product quality (color, taste, smell, interior and/or exterior appearance) was rated as very important, accounting for 70% of the total number of responses. The region/area of origin criterion was rated as important by 36% of respondents, the manufacturer's logo by 42%, the Protected Geographical Indication by 33%, the inscription on the product by 41%, the familiarity with the product by 50%, the type of dishes with which respondents want to consume it by 36%, the attractiveness of the packaging by 36% and the price of the product by 42%. From the information presented, it appears that when choosing a wine product, the majority of the criteria listed were given the important qualifier.

Analyzing the answer to the 13th question regarding the frequency of consumption of wine products, we recall the tradition of rural natives to grow vines and produce wine for personal consumption with family and friends. From 106 answers obtained, it was found that wine products are consumed daily by 1 person or 0.9%, weekly by 4 persons or 3.8%, monthly by 8 persons or 7.5%, on holidays (religious, national, family) by 70 persons or 66%, when meeting with friends and loved ones by 59 persons or 55.7%, and randomly (only when I want to consume wine products) by 26 persons or 24.5%. The exposed results can be seen in Figure 4.

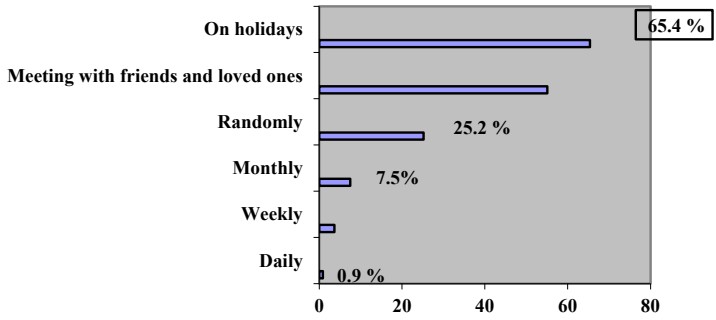

**Figure 4.** The frequency of consumption of wine products.

Questions 14–20 of the questionnaire, as previously mentioned, refer to wine tourism and for this reason, we will examine them more thoroughly, being objects of particular interest to the authors of this scientific research.

On the basis of the 14th question in the questionnaire, it was determined how many of the respondents surveyed had participated in organized tours of wine enterprises in the Republic of Moldova. This question was answered by 110 respondents.

Of those interviewed, 42 people or 37.5% visited only one winery, 42 people or 37.5% visited several wineries and 28 people or 25% did not visit any winery. The exposed results can be seen in Table 6.

**Table 6.** Share of visits to wineries.

| Characteristics of Wine Visits | Percentages |
| --- | --- |
| The share of respondents who visited a single winery | 25 |
| The share of respondents who visited several wineries | 37.5 |
| The share of respondents who did not visit any winery | 37.5 |

As a result, we conclude that at least 75% of the total number of respondents surveyed had visited wineries.

The next topic, researched in question 15, concerned the method of organizing the visits to wineries in the Republic of Moldova. 87 people answered this question. Thus, 68.2% or 60 persons visited the wineries on their own, 14.8% or 13 persons visited the wineries on visits organized by the tour company or tour operator, and 17% or 15 persons visited the wineries organized in combination (on their own and by the tour company or tour operator). The exposed results can be seen in Table 7.

**Table 7.** The organization of visits to wineries.

| Characteristics of Wine Tourists | Percentages |
| --- | --- |
| The share of tourists who visited wineries on their own | 68.2 |
| The share of tourists who visited wineries through tour operators | 14.8 |
| The share of tourists who visited wineries in a combined way | 17 |

The 16th multiple-choice grid question was tasked with researching what motivates visitors to visit a winery. A total of 109 respondents answered this question. The surveyed respondents went to wineries to visit a winemaking cellar, underground city or uniquely designed halls totalled 39 people or 35.8%, for wine tasting 28 people or 25.7%, to purchase wines and souvenirs at wine shops next to the factory 20 people or 18.3%, to dine at the restaurant next to the winery 20 persons or 18.3%, for accommodation in a rural area 11 persons or 10.1%, for additional activities offered by the wineries (boating, cycling, swimming pool, sauna, etc.) 21 persons or 19.3%, for the tourist attractions in the area proposed for visiting by the tourist agency 31 people or 28.4%; 36 people or 33% listed all the activities. The exposed results can be seen in Figure 5.

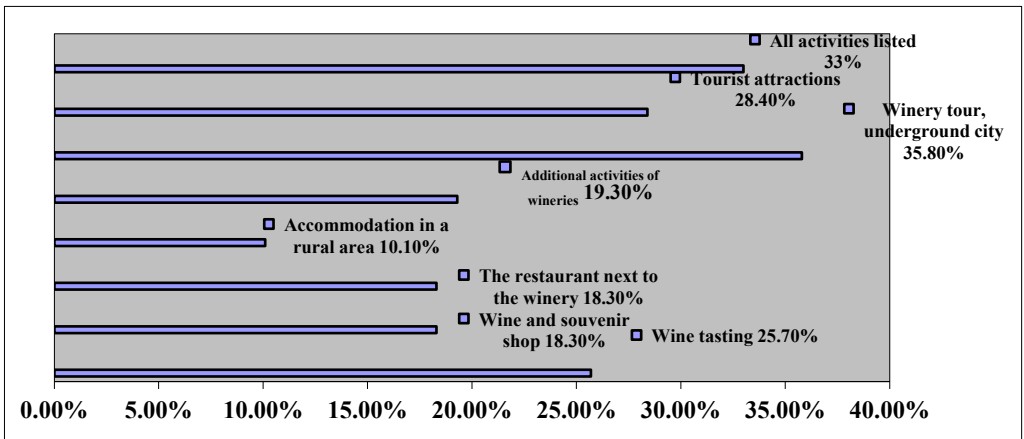

**Figure 5.** The composition of the tourist product of the wineries.

The 17th topic researched in the questionnaire was the name of the wineries visited by the respondents. Out of the total number of participants in the survey, 92 people visited wineries. Of these, the Cricova winery was visited by 57 people or 62.6% of the respondents, Milestii Mici was visited by 27 people or 29.7% of the respondents, the MIMI castle was visited by 20 people or 22% of the respondents, Chateau Purcari was visited by 16 people or 17.6% of the respondents, Chateau Vartely was visited by 22 people or 24.2% of the respondents, Asconi was visited by 23 people or 25.3% of the respondents, Poiana was visited by 9 people or 9.9% of respondents, other wineries were visited by 13 people or 14.3% of the respondents. The exposed results can be seen in Figure 6.

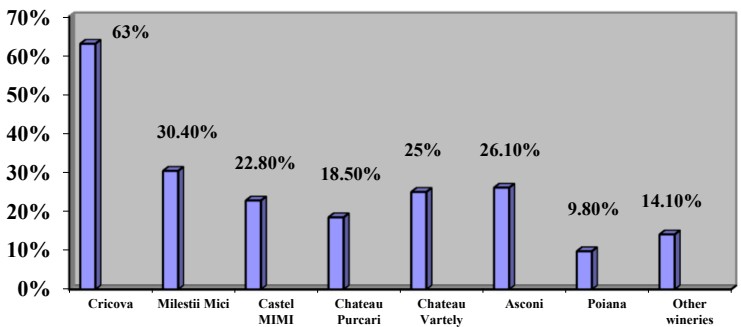

**Figure 6.** The number of visits of the respondents to the most popular wineries in the Republic of Moldova.

The 18th question in the questionnaire related to the intention to visit other wineries in the Republic of Moldova. Out of 106 answers, we find that 50 persons or 47.2% will visit wineries again, 34 persons or 20.8% will possibly visit wineries and 22 persons or 32.0% do not currently intend to visit wineries.

Therefore, we conclude that, out of the total number of respondents, 68% will or will consider further visiting the wineries. This result proves that wine tourism is popular among the population of the Republic of Moldova and has a tendency towards expansion.

The 19th question in the questionnaire asked about the level of satisfaction of visitors to the wineries. 91 people answered this question. The results of the survey show that 43 people or 47.3% were highly satisfied, 41 people or 45.1% were satisfied, 4 people or 4.4% were moderately satisfied, 1 person or 1.1% was less satisfied and 2 people or 2.2% were dissatisfied. The exposed results can be seen in Figure 7.

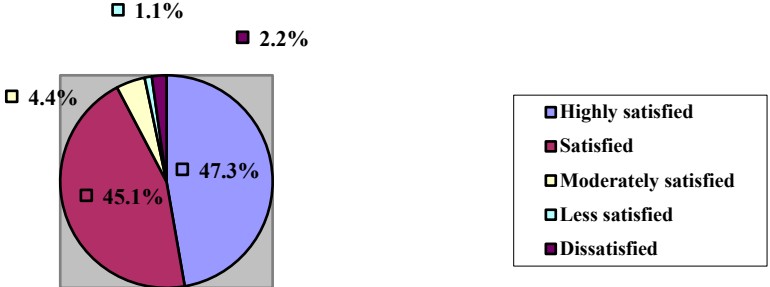

**Figure 7.** The level of satisfaction of visits to the wineries.

The 20th research topic was wine consumption education. Respondents were asked whether there is a need for wine education and whether this aspect is widely promoted by wineries. 86 people or 79.6% of respondents answered yes to this question, 19 people or 17.6% of respondents answered sometimes and 3 people or 2.8% of respondents answered no. From what has been reported, it can be concluded that a task of wine tourism is to promote wine consumption and educate consumers about the type of wine consumed depending on the dishes served [61–65].

## 5. Conclusions

Winemaking by Moldovans has been a craft from ancient times. The majority of rural households grow, produce, and serve wine. As a rule, every Moldovan event is accompanied by the serving of wine products. In the last 20 years, wine tours have become increasingly popular. As a result of going on wine tours, tourists have the opportunity to discover the beauty and richness of the centuries-old traditions of the winemaking process. At the same time, they can combine the pleasure of wine tasting with the discovery of unique itineraries and locations, with all their historical and social significance, in an exceptional natural setting such as monasteries, picturesque places, tourist attractions, rural households, workshops of folk craftsmen and so on.

The present research was carried out on the basis of a survey in order to discover the preferences of different wine consumers, to elucidate the interest of the population that participated in the survey towards trips to wineries in the Republic of Moldova, and to determine their degree of satisfaction as a result of these visits.

We believe that the opinions obtained as a result of the conducted survey can give us a fairly truthful picture of the researched questions, since respondents of different ages and levels of study participated in the survey (Table 5).

Let us establish the main conclusions and recommendations:

- Of the total number of respondents surveyed, 91.8% are consumers of wine products. Therefore, the answers obtained from the survey will allow us to discover to a fairly good degree the views of the respondents to the points covered by the questionnaire.
- Many information sources/channels (media (television, press, etc.), internet, store shelves, specialized stores, national days, festivals, exhibitions, friends/colleagues) regarding wine products are available to consumers. So the information needed by consumers can easily be found.
- Most often, wine consumers purchase wine products in specialized stores and in markets.

- Patriotism, habits passed on by parents and grandparents, recently acquired habit, suggestion of another person, simple curiosity, promotion through different channels, food safety, product characteristics (appearance, exterior, color, smell, taste, etc.), curative benefits (beneficial to health in small quantities) are important and very important reasons when wine products are chosen for consumption (Table 5).
- Important criteria in choosing a wine product include: region/area of origin, producer's logo, Protected Geographical Marking inscription about the product, familiarity with the product, type of dishes with which it is consumed, quality of the product (color, taste, smell, interior and/or exterior appearance), attractiveness of the packaging, price of the product (Table 5).
- The surveyed respondents most often consume wine products on holidays (religious, national, family) and when meeting friends and loved ones.
- Of all respondents interviewed, 75% had visited a winery and 25% had not. This fact reveals the interest of natives in visiting wineries.
- Visits to wineries were undertaken directly by 60 people or 68.2%, on the basis of combined own initiative and tour operator arrangements by 15 people or 17%, and organized by a tour operator by 13 people or 14.8%. Therefore, the majority of natives' book tours by phone or online.
- The purpose of the visit to the wineries for the surveyed respondents to a greater or lesser extent was to visit wineries, underground cities, uniquely designed rooms, tasting some wine products, purchasing wines and souvenirs, having a meal at the restaurant next to the winery, staying in rural space, additional activities offered by the wineries (boating, cycling, swimming pool, sauna, etc.), for the tourist attractions in the area, etc.
- Visiting wineries by Moldovans has become popular. The surveyed respondents visited all the wineries from personal choice. At the same time, we may mention that the most visited wineries are: Cricova and Milestii Mici.
- 68% of the surveyed respondents intend to visit the wineries again or to consider doing so. This result shows that wine tourism is popular for domestic tourists in the Republic of Moldova and has an increased interest among the population of the country.
- 47.3% were very satisfied with the visits to the wineries, and 45.1% of the respondents were satisfied. Therefore, the expectations of the surveyed visitors are justified to the extent of 92.4%.
- 97.2% of respondents believe that wine consumption education is necessary and this is a task for wine companies in the process of guiding organized tours. Therefore, the speeches presented by winery guides should contain educational information related to wines and wine consumption.

The most important factor that contributes to stimulating the consumption of wine products and wine tourism in the Republic of Moldova, according to this research, is represented by the preferred sensory hypothesis, which is followed by the hypothesis of the consumer's attitude. These factors should be analyzed in a future research study and capitalized on within a strategy for the development of wine tourism in the Republic of Moldova.

The fuzzy methodology for analyzing the importance of the factors that influence the behavior of consumers of wine products and the development of wine tourism in the Republic of Moldova is limited by the influence that manifests itself over time with changing consumption styles against the background of intercultural influences. To eliminate these limitations, the development of a neutrosophic fuzzy model is required in a future research study that will be dedicated to the analysis of intercultural influences in the Republic of Moldova.

All the results obtained tell us that in the Republic of Moldova both wine products and wine tourism are viable and popular, and the country's winemakers must constantly improve wine and wine tourism products to satisfy the needs of each consumer.

Although the present research on consumer perception of wine tourism offers an image of its potential, the current policies to support the development of wine tourism

need the involvement of institutions and local administrations to support the formation of consortia for the development of projects such as wine roads. Of course, this is difficult to organize, but without the collaboration of all the actors, the wine roads remain abandoned because the consortia are not in a position to give directions and the local administrations are not actively engaged with enhancing the heritage of wine tourism.

**Author Contributions:** Conceptualization, V.G., D.B. and G.A.Z. investigation, V.G., D.B. and L.Ș.; methodology, G.A.Z. and C.L.Z.; software, G.A.Z.; validation, G.A.Z.; writing—original draft, V.G. and D.B.; writing—review and editing, C.L.Z. and L.Ș.; supervision, G.A.Z. All authors have read and agreed to the published version of the manuscript.

**Funding:** This research received no external funding.

**Institutional Review Board Statement:** Not applicable.

**Informed Consent Statement:** Not applicable.

**Data Availability Statement:** Data is contained within the article.

**Conflicts of Interest:** The authors declare no conflict of interest.

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
