# Peer review of "Research on Consumer Perception Regarding Wine Products and Wine Tourism in the Republic of Moldova"

_agriculture, doi:10.3390/agriculture13030729_

Round 1
Reviewer 1 Report
Dear authors,
thank you for the opportunity to review your manuscript on consumer preferences and wine tourism in Maldovia. An overall, interesting work, but maybe some further work and my comments will help to improve the manuscript in its present stage.
1) Introduction:
I get the overall message of the introduction but can the authors back up the work with statistical information, industry or retail information to hilight the importance of wine consumption and toursim in Maldovia. In its present stage that is missing.
Can the authors elaborate on wines and styles ( old world/ new world wines), preset in Maldovia,
Further could the authors please indicate where within the recent body of literature their work is grounded. This shall allow to emphasize the value and merit of this work
The paragraph before the aim of the study needs to be revised. Sorry for the frankness but this reads like a copy over from an industry report or a fund application. Please consider my previous comment and revisit. Indicate the academic value of your work . Consumer preferences, wine tourism and satisfaction are well explored concepts- where do you fit in and what is new.
2) Literature review:
A literature review is completely missing. It would be expected that the state of the art on consumer preference, wine tourism and consumer satisfaction is presented. If the Maldovian literature is scant, please borrow from other countries that are similar in terms of their wine industry.
Do you have specific hypotheses?
3) Material and Methods
The first paragraph of the section suggests that a mixed method approach for the study was used. Can the authors specify what type, explain and justify the appropiateness of their choices.
Can the authors describe the survey in more detail. Items, scales and from which studies these stem from?
Can the authors explain how the 110 people were practically recruited. And how many people you had recruit to get to 110. Dropout rate, other reason for exclusion. Where people compensated for participation?
Please provide more details about the semis-structured interviews. Back up with referencing
Results and discussion
Figure 2: Sample size missing. Currently the graphic says nothing
Overall, can the socio-demographic section be presented in a table and presented in a more concise manner
Including sub-chapters would guide the reader through the manuscript better. In addition comparison with the literature is largely missing
Conclusion
Can the authors critically reflect on their work and add a limitation section--> Research approach and sampling.
Can the authors make suggestions for future research
The authors need to double check their referencing and be attentive the language appears very colloquial at times.
Author Response
- I would complete the introduction with references to the importance of wine tourism in the Czech world and the new world. At the end of the introduction, I presented the purpose and structure and the new elements of this research.
- 2. I introduced the "Literature review" section in which I introduced references to how the researched issue is approached in European countries with a tradition in the wine sector.
- 3. I completed the methodology with references to the data on how we conducted the survey and with a schematic structure of the main analyzed hypotheses. Also, the sociodemographic data were entered in the form of tables.
- 4. I completed the conclusions section with references to the limits of the model used in the analysis and the opportunities for future research.

Reviewer 2 Report
I would like to thank the authors for a very interesting research topic, but I would like to make some corrections in order to improve the quality of the manuscript.
The title is clear and indicates the entire research. The abstract contains all the necessary elements.
I would suggest that the review of literature be strengthened, so that the slots will insert literature related to similar problems and researches for each of the variables they investigated.
The methodology is explained precisely.
I suggest inserting research flow or design of hypotheses in the form of a figure, in order to give the readers a clearer picture of the research.
Also, figures describing demographic characteristics can be tabulated, not pie. Given that there are no specific statistical methodologies for examining differences or predictions, only a description, it may be better to present the results in tables. I ask the authors to present the hypotheses and later confirm the hypotheses.
In the results section, add a discussion in the sense that the research and results are related to similar research on the given topic.
Expand concluding considerations, add limiting circumstances, future theoretical and applied implications.
Expand references.
Gajić, T., Đoković, F., Blešić, I., Petrović, M.D., Radovanović, M., Vukolić, D., Mandarić, M., Dašić, G., ; Syromiatnikova, J.A., Mićović, A. (2023). Pandemic Boosts Prospects for Recovery of Rural Tourism in Serbia. Land, 12(3), 624. DOI: 10.3390/land12030624
Gajic, T., Raljic, J.P., Blešic, I., Aleksic, M., Petrovic, M.D., Radovanovic, M.M., Vukovic, D.B., Sikimic, V., Pivac, T., Kostic, M., et al. Factors That Influence Sustainable Selection and Reselection Intentions Regarding Soluble/Instant Coffee—The Case of Serbian Consumers. Sustainability 2022, 14, 10701. https://doi.org/10.3390/ su141710701
Author Response
I introduced a section entitled literature review
I changed the representation of the demographic characteristics from the pie chart in the table
I have inserted a figure with the projection of the hypotheses in the form of a figure and also comparative analysis of the confirmed hypotheses in the result section.
I have added in the results section a discussion about similar research on the topic of wine tourism.
I introduced in the conclusions references to the limits of this research and to future application developments.
I have expanded the references with the proposed recommendations.

Reviewer 3 Report
Reviewer coment:
- The title corresponds to the content of the paper. +
- This study represents significant contribution for the development of tourism on the base of tradition of vine production in Moldova with respect of citizens experience and consumer estimation.
- The main question of paper addressed to study development of the tourism on the base investment in winery and supporting wine owner to create new quality wine products.
- The aim of research is not clearly and fully pointed out on the end of chapter of Introduction
- The rule is that aim of study need write on the end of chapter of introduction.
- Should be pointed out aim of investigation at the end of Chapter of introduction. *
- Key words are appropriate. +
- Scientific methodology is applied correctly for this type of study. +
- Results are clearly presented and discussed.
- Tables, figures, pictures are clear.+
- Conclusions need improve. conclusions should be made only on the basis of research results. In chapter Conclusion is not allowed quote refrences..*
- This study represents complementary to the previous ones. +
Reviewer coment:
In line 90 . …..” relation to to the …- delete one to
Author Response
I introduced at the end of the introduction a presentation of the purpose of the work.
In line 90 I made the corrections.
I removed the references from the Conclusions section
Reviewer 4 Report
Dear authors:
Emphasize that the subject it deals with is of interest to the scientific community. There are many authors who propose case studies on the importance of wine tourism as an element that can generate wealth and an element that allows the development of territories.
1.- The introduction lacks a development structure for the topic that highlights the importance of wine tourism in Europe, some data should have been given and also from Europe. And of course, express the structure of the research, clearly defining the objectives of the study and leaving the scope of study geographically expressed on a map. The introduction does not respond adequately to the approach of a level academic investigation.
2. Citations and authors are missing. There is no literature review section, which allows observing the field of research of other researchers who deal with these issues. Especially from countries like Italy, France or Spain.
4. The methodology is deficient because a clear technical sheet is not presented so that the reader observes what has been done and how it has been done, what degree of reliability, etc. This is a major flaw in this investigation, which could be corrected by using tables and/or figures clearly show the method followed.
5. The results are also presented in an unclear way, although the interest of the results is intuited, they could also be improved. Perhaps if the survey were annexed it would be much easier to check the results
6. A discussion section is missing to complement the conclusions, which by the way, really as they are written seem "more results." In this sense, an in-depth review of the conclusions is needed to really allow us to observe what conclusions are reached.
In general, a low level is observed in this article as it is presented, although it could be corrected with a deep rewrite and citations from other authors, etc...
Best regards
Author Response
- In the introduction I introduced references to the importance of wine tourism in Europe. At the end of the introduction, I presented the purpose and structure of the research.
- I introduced the "Literature review" section in which I introduced references to how the researched issue is approached in European countries with a tradition in the wine sector.
- We completed the methodology with a schematic structure of the main hypotheses analyzed.
- We improved the presentation of the degree of importance of the hypotheses analyzed by means of the fuzzy model developed in this sense
- I rewrote the article according to the suggestions, and completed it with citations.

Round 2
Reviewer 1 Report
All my comments have been considered and implemented. Thank you!
Author Response
Thank you for your support.
Reviewer 4 Report
Dear authors,
The revision that has been made of the text is profound and now it is easier to follow its reading.
1.- The introduction has been modified and there is now a starting point for the investigation that reveals the interest of the topic. I suggest that you put a reference map of the place where the investigation is focused. The reading of this magazine covers the international scope and this should be taken into account.
In the introduction, when talking about the importance of wine routes in certain countries, there is no quote from various authors that reflects that the subject is mastered at an international level.
2.- The methodology is much better now, although you could look at other works where the characteristics of the surveys are reflected in a table and at a glance you can see what has been analyzed and its scope. I suggest this be done.
3.- The contributions made to results have improved considerably
4. The results are better defined although their own nature is not obligatory, that is, they would not have to refer in the text to the tables where the conclusion comes from, this could be avoided.
5. Finally, the new presentation of this article and the corrections made should be kept in mind in the summary, which should now more clearly include the objectives, the methodology used and some better specified idea of results.
Good luck
Author Response
In Introduction section I insert a map of wineries in Moldova on which the research of consumer perception regarding wine products and wine tourism is concentrated. I also completed the introduction with bibliographic references about the wine route.
I made changes and I have represented the characteristics of the survey in a table. I also formulated the conclusions in a synthetic form.